# Dihydroxyacetone: A User Guide for a Challenging Bio-Based Synthon

**DOI:** 10.3390/molecules28062724

**Published:** 2023-03-17

**Authors:** Léo Bricotte, Kamel Chougrani, Valérie Alard, Vincent Ladmiral, Sylvain Caillol

**Affiliations:** 1ICGM, Université de Montpellier, CNRS, ENSCM, Montpellier, France; 2LVMH Recherche, Département Innovation Matériaux, 45800 Saint Jean de Braye, France

**Keywords:** bio-based, carbohydrates, DHA, monomer, polymer, glycerol

## Abstract

1,3-dihydroxyacetone (DHA) is an underrated bio-based synthon, with a broad range of reactivities. It is produced for the revalorization of glycerol, a major side-product of the growing biodiesel industry. The overwhelming majority of DHA produced worldwide is intended for application as a self-tanning agent in cosmetic formulations. This review provides an overview of the discovery, physical and chemical properties of DHA, and of its industrial production routes from glycerol. Microbial fermentation is the only industrial-scaled route but advances in electrooxidation and aerobic oxidation are also reported. This review focuses on the plurality of reactivities of DHA to help chemists interested in bio-based building blocks see the potential of DHA for this application. The handling of DHA is delicate as it can undergo dimerization as well as isomerization reactions in aqueous solutions at room temperature. DHA can also be involved in further side-reactions, yielding original side-products, as well as compounds of interest. If this peculiar reactivity was harnessed, DHA could help address current sustainability challenges encountered in the synthesis of speciality polymers, ranging from biocompatible polymers to innovative polymers with cutting-edge properties and improved biodegradability.

## 1. Introduction

“Until the mid-20th century, carbohydrate chemistry was considered a minor and separate branch of organic chemistry. Despite the considerable importance of carbohydrates in nutrition, food industry, biology and medicine and as organic raw materials, they were regarded by many organic chemists with some doubts. Sugars are not soluble in organic solvents, only in water; they cannot be distilled and often do not crystallize; and—worst of all—when a reducing sugar is dissolved in water, it produces a solution containing at least six different compounds. Not at all like organic compounds.” [1]. This is how Stephen J. Angyal, originally an organic chemist, was describing the place of carbohydrates in the chemistry research field, after nearly 50 years of prolific work on sugars.

1,3-dihydroxyacetone (DHA) is the simplest compound answering to the conventional definition of a carbohydrate: a compound with the chemical formula C*_m_*(H_2_O)*_n_*. It is primarily used as an ingredient in sunless tanning products. Minor other uses are reported in pharmaceutical, medical and food applications since DHA is an FDA-approved chemical [2,3,4]. Since the first papers reporting its discovery in 1860 [5], DHA has been largely overlooked even by carbohydrates researchers, with only roughly a hundred publications devoted to DHA until 2021. In contrast, glyceraldehyde, its closest isomer, and glycerol, its main precursor, were the topic of several thousand articles over the same period (Figure 1).

However, in the last few years, DHA has made some interesting appearances in various scientific journals, over a broad range of topics. For example, it has been reported to be present at the very early stages of life on Earth, as one of the earliest sugars produced by reaction of formaldehyde containing traces of glycolaldehyde heated in calcium hydroxide solution [6,7]. It has also been reported as a major component of e-cigarette smokes and was proven to react with DNA in vitro [8,9]. This close relationship with biochemical processes in living organisms makes DHA a well-known, yet seldom studied chemical. DHA is, from a broader perspective, foreseen as a valuable waste-valorization compound from other miscellaneous industrial processes, as the market demand is growing quickly and expected to reach USD 250 million by 2030 [10]. As an example, Rumayor et al. performed a feasibility study dealing with an industrial CO_2_-recycling plant that would reject DHA as a waste by-product [11].

To our knowledge, in the recent literature, reviews about DHA have either focused on DHA derivatives for fine chemistry [12], on DHA bioeconomics [13], or on the use of chemically protected forms of DHA for the synthesis of linear polycarbonates [14]. In the following, we review the literature concerning DHA as a building block for bio-based polymers. The first section deals with the discovery of DHA, its role as a metabolite in living organisms, its industrial production and applications. The second section focuses on the range of reactivities of DHA, detailing further the various products involved and the experimental conditions. The second section concludes with a review of the literature dealing with the synthesis of DHA-based polymers and provides the reader with an overview of the emerging potential of DHA as a bio-based building block for materials.

## 2. An Overview of DHA, from Key Metabolite to Sunless Tanning

### 2.1. A Key Step in the Glycolysis Process

As mentioned previously, DHA is known to be closely involved in the biochemistry of life. It is a crucial link in the glycolysis process, a metabolism step that is traced even in the oldest forms of living organisms several million years ago [15]. In glycolysis, glucose is first transformed into glucose-6-phosphate and then into fructose-6 phosphate and fructose-1,6 bisphosphate by several enzymes. At this point, the action of an aldolase on fructose-1,6 bisphosphate splits the molecule into two isomers: dihydroxyacetone phosphate (DHAP) and glyceraldehyde phosphate (GADP). These compounds, which have the ability to interconvert into each other under the catalytic action of triosephosphate isomerase, play distinct roles: only GADP is able to be processed further in the glycolysis process (Figure 2).

Triosephosphate isomerase takes on the role of regulator of this step of glycolysis, enabling control of the amount of GADP flowing in the second half of the glycolysis process. On the other hand, DHAP, as the more thermodynamically stable isomer, is used as a stock waiting to supply the metabolic process when needed and is, thus, further processed into triglycerides when in excess. DHA’s phosphated derivative, thus, plays a key role in metabolism.

### 2.2. Discovery of DHA’s Chemical Properties and the Maillard Reaction

The discovery of the chemical properties of DHA is what is called a serendipitous finding, as it was as interesting and stunning as it was unexpected. DHA was first used as a medicine to treat diabetic patients in the 1930s, but the scientists running the tests were surprised to observe a strong and unusual yellow staining of the patients’ gums [16]. A first artificial suntanning lotion was then released in California in 1945, but the coloration obtained was an unnatural orange color with strong stains on palms and other zones where the skin’s epidermis is thicker due to the lack of recommendations for an application method.

A few years later the first reported tests of the staining effect on skin of DHA were run by E. Wittgenstein and H. K. Berry and presented in a *Science* paper in 1960 [17]. Although they reported the previous interest of their research community in the skin-staining effect of DHA, they were the first to identify significant elements to explain the underlying reaction mechanism. The study came with convincing evidence that DHA staining proceeds through combination with amino groups in skin proteins. After this seminal paper, the reaction process was studied further and the coloration was attributed to a closely related process, which occurs by chain chemical reactions between amino acids and reducing sugars, discovered a few decades ago by the French chemist Louis Camille Maillard (Figure 3) [18,19,20].

The Maillard reaction is known to occur in food and is responsible for the brown coloration and grilled flavor in bread and steaks, among others. Along with the caramelization process, which involves a partial pyrolysis of sugars, Maillard reactions are the main pathways to achieve non-enzymatic browning reactions. Upon heating, the carbonyl compounds of reducing sugars proceed to a condensation reaction on the primary amino groups of the amino acids producing an unstable *N*-substituted glycosylamine (a Schiff-base compound bearing a glycosyl group) and water. This glycosamine compound then undergoes an isomerization known as Amadori or Heyns rearrangement depending on the nature of the sugar, which leads either to aldosamine or ketosamine compounds [22]. Once this first step is over, a long and complex series of chemical and structural rearrangements starts, the scientific comprehension of which decreases quickly along the process. The chain-reaction usually proceeds and results in a complex mixture of brown chromogenic, cyclic polymeric compounds, named melanoidins. The structure of these polymers is close to that of melanin, the range of chromophores produced by human skin upon prolonged exposure to sunlight, commonly named tanning. However, it is noteworthy that melanoidins only provide a very slight protection to UV radiation unlike the skin’s melanin, and, therefore, artificial suntanning using a DHA-based lotion should always be accompanied by the use of a suitable sunscreen upon sunlight exposure.

### 2.3. Industrial Production and Further Uses of DHA

As mentioned above, DHA is primarily used as an ingredient in sunless tanning products, although minor uses are reported in various other fields (Figure 4). It has been evaluated as a potential medicine for several diseases: for children with glycogen storage disease (diabetes) [14,17,23,24,25,26]; as a complementary photoprotector for patients suffering from various porphyric diseases, which are liver disorders leading notably to increased skin photosensitivity [27,28]; as an antifungal agent to cure dermatomycosis afflictions [29,30]; or as a symptomatic treatment for psoriasis [31] and vitiligo, a skin disorder resulting in the appearance of depigmented stains on the skin of affected patients [13,16,28,32]. DHA is also used as an important feedstock for the production of a range of industrially important chemicals, such as 1,2-propylene glycol, lactic acid, methotrexate, DL-serine and various surfactants [30,33,34,35,36]. Finally, DHA has also been assessed as an ingredient for controlled release mosquito-repellent formulations [14,31,32,37,38,39], as a muscle endurance enhancer in diet formulations [14,40], and as a preventative for cyanide poisoning [14,41,42].

DHA is widely available as it is almost exclusively obtained by mild oxidation of glycerol, which is a major byproduct of the industrial production of biodiesel (1:10). With the increasing production of biodiesel, the market has been flooded with glycerol, resulting in an almost negative value of the crude glycerol and a very low value of purified glycerol (∼0.6 US$/kg in 2018) [43]. Therefore, finding ways to valorize this byproduct became urgent, and DHA is one of the best choices available considering its high value (∼150 US$/kg) [43].

The mild oxidation of DHA can be performed following three main pathways, as reported in the literature: by chemical oxidation in aqueous solution, by electrooxidation, or through microbial fermentation (Figure 5).

The chemical oxidation route of glycerol to DHA has been explored in basic aqueous solution, with a range of oxidizing agents such as benzoquinone (BQ), H_2_O_2_, Cr-based reagents or O_2_, that can be pressurized or not [44]. The first issue has been that, using common catalysts for alcohol oxidation, such as Au, Pt and Pd, yielded mainly glycerate, the conjugate base of glyceric acid. It resulted in a favored oxidation of the primary alcohols of glycerol, whereas DHA would be obtained by oxidation of the centered, secondary alcohol. The process has then been improved by use of Au nanoparticles on metallic oxide supports (Al_2_O_3_, ZrO_2_, TiO_2_, NiO, ZnO and CuO) in base-free aqueous solutions, reaching 59.6% selectivity of DHA for 86.4% glycerol conversion, and up to 95.6% selectivity in DHA for 72.7% glycerol conversion [44,45,46,47,48]. Although these extensive studies are interesting for increasing the conversion yields of this process, the use of a catalyst implies high costs and catalyst deactivation issues. Indeed, a 2011 study estimated that the catalyst represented 55–86% of the production cost of this reaction, and the reported yields were still relatively low [49]. For these reasons the selective oxidation of glycerol to DHA in aqueous solutions was still at laboratory scale in 2020 and far from reaching commercial application [50]. Other sporadic attempts have been made in organic solvents [51] or by gas-phase oxidation [52], reaching yields up to 92% and 90% respectively, but the economic viabilities of these processes have not been evaluated to our knowledge. A patent involving the use of Lewis acids combined with specific chlorine derivatives as sources of ClO^+^ has also been reported [53,54].

Electrooxidation (EO) of glycerol to selectively yield DHA has attracted a lot of interest regarding the understanding of the chemical oxidation pathway. EO is a chemical process involving an aqueous solution through which an electric current is applied and results in redox reactions both at the anode and cathode of the electrooxidation apparatus by formation of highly reactive species [55,56,57]. It has recently gained interest mainly for wastewater treatment, where the absence of a need for further reactants is a major benefit [57]. The electrodes are usually made of (or at least coated with) noble metals called electrocatalysts, the nature of which affects the efficiency and selectivity of the occurring reactions. However, due to the high costs and strategic aspects of these metals, ongoing research is being undertaken to look for innovative electrocatalysts, such as non-noble metals or non-metal (N, P, B, S) doped hybrid nanocatalysts, among others [30,58,59]. A crucial issue is the search for an electrocatalyst that would be both cost effective and sustainable [30]. Numerous recently published papers evidence a strong interest in this new way to achieve DHA synthesis for glycerol, through use of innovative catalysts [60,61,62,63,64,65,66]. However, the low glycerol concentration used, the lack of studies of the continuous flow process, and still relatively low yields and/or selectivity account for why no pilot scale studies have been reported so far [67,68].

At the industrial scale, DHA has been produced for decades by microbial fermentation. Microbial processes in general allow access to a range of reactions that may be either cost-inefficient or yield-inefficient if conventional chemical processes were used instead [34]. DHA production by microbial fermentation is a typical example of this situation. The standard process takes place in a large, aerated reactor, filled with a starting glycerol aqueous solution inoculated with a microbial pre-culture and stirred for the reaction to proceed with glycerol feeding. However, both DHA and glycerol increasingly and irreversibly inhibit the reaction, so their concentration must be kept below a critical threshold in the reactor to preserve cell viability [34]. This concern makes the process particularly onerous to run, as it also needs cleaning and sterilization between batches. Yet, the process is still the only industrialized route for production of DHA in the world.

In this sense, process optimization and innovation of the fermentation of waste glycerol toward DHA has been extensively studied. This work has been recently reviewed by Ripoll and Betancor, who reported up to 65.05% yield of DHA, using advanced techniques, such as fed-batch reactors with initial glycerol concentration of 10–30 g/L over 156 h [69,70]. Alternatively, they report 61.9% DHA yield for batch reactors over 138 h, with selected sources of bacteria facilitating increase in the glycerol initial concentration up to 60 g/L [69,70]. The authors highlight that there is “still plenty of room for improvement”, as many parameters are yet to be optimized [69].

The recent advances in the three routes for DHA synthesis are encouraging and the scientific community is heavily engaged in the development of greener processes. Researchers working on the chemical oxidation of glycerol are increasingly looking for sustainable catalysts which can achieve the synthesis using either O_2_ or H_2_O_2_ as reactants, instead of more polluting and/or fossil-based reactants. Electrooxidation is emerging as a sustainable process, as long as the electricity used is produced from renewable sources, in aqueous solution and without any additional reactants. Finally, the production of DHA through microbial fermentation still has a lot to offer, with plenty of advances in process optimization and potential to improve the yield of this route using cutting-edge applied technology, such as the genetic modification of biocatalysts or metabolic engineering, among others [69].

## 3. A Multi-Functional Synthesis Tool: DHA’s Broad Range of Reactivities

As previously mentioned, the chemical formula of 1,3-dihydroxyacetone is C_3_H_6_O_3_, which makes it one of the two simplest compounds, along with glyceraldehyde, to be considered as carbohydrates (C*_m_*(H_2_O)*_n_*).

DHA is a highly functional, yet quite simple, molecule, with a broad range of potential reactions (Figure 6).

DHA is an isomer of glyceraldehyde (Gld), and, together, they are generally referred to as “trioses”, meaning three-carbon monosaccharides. The L- and D-glyceraldehyde enantiomers are termed “aldoses”, whereas DHA is named a “ketose” since it possesses a ketone function. As carbohydrates, they can undergo a very wide range of well-studied, but often not yet fully understood, reactions, each of which will be described below (Figure 7).

### 3.1. DHA—Gld Interconversion: The Lobry de Bruyn—Alberda van Ekenstein Transformation of Carbohydrates

An interesting property of carbohydrates is their isomerization; they are able to transform from ketoses into aldoses and inversely. After decades of serious doubt regarding the mechanism of this transformation, it now seems accepted by researchers that it proceeds through a tautomeric enediol as the reaction intermediate. This reaction equilibrium is named the “Lobry De Bruyn—Alberda Van Ekenstein transformation of carbohydrates” and will be abbreviated as the LdB–AvE transformation in the following.

DHA has been used as a model molecule to study this transformation because, as one of the simplest carbohydrates, it is also subject to all the side-reactions potentially occurring during the isomerization. Efforts have since been focused on the extension of these model studies to heavier sugars, such as glucose, from the simple model of the DHA–Gld LdB–AvE transformation [71].

The LdB–AvE transformation was discovered by Lobry de Bruyn and Alberda van Ekenstein in 1895 when running experiments on simple sugars, such as glucose, fructose and mannose [72]. They observed that these sugars, under the influence of even very dilute alkalis, showed a significant change in rotational power, and established that this was due to the interconversion of these species into each other.

From there, a lot of work has been undertaken, with some contradictory theories proposed, making the state of the art about the LdB–AvE transformation somewhat unclear. Two well-documented reviews by Speck in 1958 and by Angyal in 2001 have shed light on the fact that the LdB–AvE transformation is actually a combination of two chemical mechanisms occurring simultaneously: epimerization and aldose—ketose interconversion [1,71].

Epimerization is a chemical process involving at least two diastereoisomers that are converted into each other through several possible mechanisms, depending on the structure of the compound and the experimental conditions. Since DHA is a molecule without any chiral center, it is not subject to epimerization. The aldose–ketose interconversion is, thus, the only relevant reaction for DHA.

LdB–AvE is generally accepted to proceed via an enediol anion intermediate, but this assumption has been controversial for decades after the proposal of this hypothesis [73]. Even though several studies have observed hydrogen exchange with the solvent when deuterium- and tritium-labeled sugars or deuterated water were used [74,75,76,77], numerous papers have suggested modifications to the original enediol path. One of these suggested alternative paths to the enediol-mediated LdB–AvE transformation is supported by more evidence than the others and has been accepted by carbohydrate chemists. A small proportion of the product formed during LdB–AvE involves a cation-catalyzed epimerization mechanism, especially when the reaction is carried out in the presence of calcium complexes [78,79,80,81]. However, this reaction seems to be limited to glucose/mannose and xylose/lyxose systems since DHA is a molecule without any chiral center and does not appear to be transposable to other systems including DHA–Gld.

The few studies that concern the DHA–Gld interconversion have taken place under slightly different conditions. Indeed, in addition to the fact that the DHA–Gld interconversion has been observed even in the absence of a catalytic base, it has also been observed that the enediol intermediate is in its undissociated form, and not anionic, as for LdB–AvE transformations of other sugars [1]. This form appears to be stabilized by an intramolecular hydrogen bond, connecting the hydroxyl groups on positions 1 and 3 and, hence, locking the enediol into a trans configuration. With these elements, the enediol of DHA is assumed to be stabilized in a pseudo-cyclic form of lower free energy (Figure 8).

These elements were provided by Yaylayan et al., who ran an extensive investigation of the DHA–Gld interconversion by FTIR spectroscopy [82]. They found that a good means to catalyze this reaction was to use a base, such as triethylamine (TEA), as a solvent, to easily reach equilibrium in a short amount of time at room temperature, allowing enough time for observation before the enediol intermediate quickly transforms into other byproducts (vide infra). Anhydrous or slightly moist pyridine has also been reported as a very effective catalyst for the DHA–Gld interconversion [71,83,84]. These experiments led to the observation that, at equilibrium, the mixture was composed of 60–65% DHA and 35–40% glyceraldehyde regardless of the starting point (pure DHA or pure Gld) [82]. This observation confirmed that DHA is more thermodynamically stable than Gld at room temperature (Figure 8) [82,84,85].

Regarding the reaction mechanism of this equilibrium, Nagorski and Richard proved by ^1^H NMR that, in a weakly alkaline solution, DHA–Gld isomerization occurs by both hydride and proton intramolecular transfer [86]. They also performed a kinetic study on these species and showed that the rate constants for both proton and hydride transfer mechanisms were closely matched, so that neither mechanism was favored over the other.

In summary, the core of DHA chemistry is its ability to undergo rapid acid- and base-catalyzed interconversion into glyceraldehyde, its aldehyde isomer. Most evidence suggests that this reaction occurs via an E-enediol intermediate, stabilized in a cyclic H-bonded conformation similar to that of glyceraldehyde.

### 3.2. DHA—Gld Dimerization and Behavior in Aqueous Solutions

Other findings from the work of Yaylayan et al. [82] include the discovery of two distinct forms of glyceraldehyde in solution, also stabilized by intramolecular hydrogen bonds like the enediol intermediate. Racemic glyceraldehyde is commercially available under a dimer form, with the appearance of an off-white powder, which dissolves quickly in water at 25 °C. Upon dissolution, Gld is present in two distinct forms in variable proportions. Both forms involve the carbonyl of the aldehyde as a proton acceptor in the H-bond, but there are two different hydroxyls, which act as proton donors, leading to two different possible conformations (Figure 9).

DHA also shows similar behavior; however, the most stable conformation for this molecule appears to be more of a 5-membered ring double conformer bearing two intramolecular hydrogen bonds [87]. DHA is also commercially available as a crystalline dimer, but is both much cheaper than glyceraldehyde and bio-based and more sustainably produced through industrial fermentation processes. It is possible to isolate DHA in its monomer form upon fresh recrystallization, but studies have shown that, upon storage even in dry conditions, it slowly reverts to the dimeric form (Figure 10) [88,89]. This means that commercially available DHA, even when it is sold as its monomeric form, will mostly contain the dimeric form. The presence of two hydroxyl groups in the alpha position of DHA’s ketone seems responsible for the formation of the dimer. Indeed, in the case of monohydroxyacetone (OH-CH_2_-(C=O)-CH_3_), the absence of one of the two hydroxyl groups led to the inability to demonstrate the existence of dimeric species of monohydroxyacetone, either in solution or undiluted [88,90].

The dimer of DHA is very soluble in water and only sparingly soluble in many common organic solvents, ranging from methanol and DMSO at room temperature, to acetonitrile and toluene with some heating. Its dissociation into the monomer form is achieved by both melting or solubilization in the presence of acids or bases as catalysts [91,92]. Some dissociation mechanisms have been proposed, and kinetic studies of this dissociation have been published [82,91,93,94,95]. The rate of dissociation was found to depend strongly on the concentration, solvent, pH and temperature [82]. In DMSO, for example, the conversion to the monomer is only 50% complete after 64 h at room temperature, whereas it takes only 20 min in water to achieve such a conversion [88].

The dimeric form of DHA has been studied by Kobayashi and Takayashi, both in the crystalline state and in solution, via FTIR, Raman and ^1^H NMR spectroscopies [96,97]. They found that DHA crystallizes in five different forms. Four out of five forms were obtained by recrystallization in methanol, and these appeared to crystallize as the stereoisomers of 2,5-bis(hydroxymethyl)-1,4-dioxane-2,5-diol (Figure 10). The fifth form was obtained by freeze-drying of an aqueous solution, and is a crystalline form of monomeric DHA, which has been observed to revert to the dimeric form upon storage. This transformation seems to be favored by exposure to air since these crystals are very hygroscopic, even transforming into an aqueous solution if no precautions have been taken [96]. An extensive study of the crystal structures of dihydroxyacetone and common derivatives is provided by Ślepokura and Lis [98].

Ketones and aldehydes are also known to undergo rapid hydration in aqueous solution [99]. For ketones, usually the hydrated form exists only in tiny amounts in solution due to the predominant stability of the non-hydrated form. For example, in aqueous solution, acetone is in equilibrium with its hydrated form propane-2,2-diol, but with a very large predominance of the ketone form (equilibrium constant K ≈ 10^−3^) [100]. In contrast, aldehydes are usually more subject to hydration. Indeed, the nature of the substituents in the alpha position of the carbonyl strongly influences the reactivity of the latter. In the case of formaldehyde, in aqueous solution, a strong shift of the equilibrium towards its hydrated form, methanediol (equilibrium constant K ≈ 10^3^), is observed [101].

Gld bears one hydroxyl group in the beta position of the carbonyl, and DHA possesses two such hydroxyl groups. These electron-withdrawing groups increase the reactivity of the carbonyl, compared to non-hydroxylated analogs, such as acetone, for example [102]. Therefore, although DHA is a ketone with a chemical structure close to that of acetone, when dissolved in water it reaches a ketone/hydrate equilibrium of 4:1, whereas hydrated acetone is present in water only in trace amounts (Figure 10) [9,88,97].

To summarize, when commercial DHA is solubilized in water, it exists in three forms in equilibrium: dimer, monomer, and monohydrate gem-diol. Since DHA is rapidly isomerized to Gld, it is also possible to find small amounts of Gld, as well as its monohydrate species. In organic solvents, hydrate formation is limited by the presence of water in the solvent, and dissociation of the DHA dimer may take significantly longer than in aqueous solution. Heating and acid catalysis may increase the rate of dissociation, but can also generate undesirable byproducts, such as those described further in this document.

### 3.3. MGO Formation and Condensation Reactions

As discussed above, the DHA–Gld interconversion is catalyzed by both acids and bases and goes through an enediol intermediate. Some studies, presented below, have shown that both DHA and the enediol, even under mild experimental conditions, tend to be subject to non-enzymatic dehydration reactions. These condensation reactions can lead to several undesirable products, either through aldol condensation of the dehydration products or by autocondensation of DHA.

The first scientist to produce pure DHA was Piloty in 1897 [103,104]. At this time, he had already described some unexpected behaviors of DHA. For example, in his original paper, he stated: “When liquid dihydroxyacetone was allowed to stand without cooling and without seeding with the crystalline material to induce crystallization, it was transformed after a long interval into a crystalline substance which was not dihydroxyacetone but a polymerization or condensation product of it, melting at 155 °C”. Then he described several other unexpected byproducts obtained, which were further investigated by Levene and Walty, when they were given a stock of DHA by a co-worker that had been forgotten in a storeroom for a year [103]. The product had turned into a sticky mass containing crystalline particles. This behavior of DHA, associated with Piloty’s observations, was attributed, after further experiments by Levene and Walty, to self-condensation of DHA, forming oligomeric derivatives of DHA by chain condensation reactions at room temperature (Figure 11). Similar observations on the autocondensation of DHA were reported by Spoehr and Strain [90].

Although they do not directly issue from DHA, a large number of side-products has also been identified as products of the dehydration of the enediol intermediate. This dehydration reaction leads to the formation of a well-known, highly reactive compound named methylglyoxal (MGO), alternatively termed pyruvaldehyde or 2-oxopropanal. MGO is the keystone of several reactions that have been observed: condensation reactions (Figure 11) and dehydration-rehydration reactions (see *Section 3.4*
*MGO Hydrates and Conversion to Lactic Acid*).

MGO formation has been mostly studied in work on carbohydrate decomposition for biomass valorization purposes [105,106,107,108]. It appears that the hexose (mainly glucose and xylose) deriving from biomass depolymerization are partly degraded to DHA and Gld via retro-aldol reactions (Figure 12).

In the specific case of DHA, its ability to spontaneously degrade to MGO has been extensively studied by a research team from New Zealand, as part of their work on the role of MGO as an antioxidant in a local honey [95,109,110,111,112]. They observed that non-enzymatic dehydration of DHA to MGO occurred even in honey, a highly viscous medium with low water availability (0.56–0.62 a_w_) and initial concentrations of DHA as low as 0.2%wt [113].

The irreversible conversion of DHA–Gld to MGO follows first-order kinetics according to the work of Grainger et al. and was reported to be catalyzed by a broad range of species. Spoehr and Strain provided evidence of catalysis by weak alkalis, including various organic amines [90,114], or a large range of polyvalent anions [90,94,115], extensively studied by Riddle and Lorenz [116]. The latter specifically showed strong catalysis by fructose and glucose phosphate species, which are known to be precursors of DHA and Gld phosphates in metabolism (Figure 2). The production of MGO from DHA–Gld was also reported to be acid-catalyzed by Lookhart and Feather, with a focus on the mechanism involved using tritium-labeled compounds [117]. Further studies showed that the reaction was actually catalyzed by both Lewis and Brønsted acids [118,119,120,121].

MGO, as a 1,2-dicarbonyl compound, is highly reactive due to its strong electrophilicity. A study by Thornalley reported α-oxoaldehydes, such as MGO, to be up to 20,000-fold more reactive than glucose in glycation process [110,122]. It is, therefore, not surprising that several studies have reported a very broad range of compounds derived from aqueous solutions containing MGO, mostly formed by aldol/retro-aldol successive condensations (Figure 11) [123,124]. Indeed, experiments on a heated, slightly acidic DHA aqueous solution have isolated products derived from MGO, often leading to various aromatic compounds, easily noticed by the characteristic color, ranging from light yellow to deep brown, they give to solutions. Such compounds were already expected by Spoehr and Strain, who tried to convert quantitatively MGO into lactic acid. They observed that the stoichiometric balance was inconsistent with MGO consumption and the production of lactic acid, suggesting that other compounds may be produced [90]. The lactic acid production path from DHA is discussed further in the next section.

To summarize, in DHA-containing solutions, one of the adverse reactions that can be expected is dehydration, as it can lead via two different pathways to undesirable by-products. Levene and Walty reported the occurrence of autocondensation of DHA, forming sparingly soluble oligomeric DHA structures. DHA was also shown to degrade by dehydration to MGO, a highly reactive intermediate that can lead to a broad range of aromatic by-products by aldol/retro-aldol condensations.

### 3.4. MGO Hydrates and Conversion to Lactic Acid

DHA has been the subject of several studies aiming to convert DHA into lactic acid (LA) (Figure 13). LA has a broad range of direct applications, making it a highly valuable byproduct for food additive, cleaning agent, pharmaceutical and cosmetics applications, among others. It is also a highly valuable bio-based intermediate for numerous commodity chemicals, such as acrylic acid, 1,2-propanediol, or acetaldehyde [125], as well as polylactide (PLA), a bio-based and biodegradable polymer mainly used for rigid packaging [126]. Morales et al. have suggested that the DHA isomerization pathway is both an ecologically and economically interesting alternative for the industrial production of LA [125]. Since the conversion of MGO to LA has been known for a long time [127], the limiting step in this process was the conversion of DHA to MGO, as shown by several studies on the kinetic parameters of this reaction [119,120,121,128]. This synthesis route is currently a trend of interest for several research teams, as interesting results have been obtained using highly efficient and recyclable Lewis-acid catalysts [118,119,120,121,128,129,130,131,132,133,134,135].

It has been suggested that LA may also exhibit a catalytic effect as a proton donor in its own production, but this effect was not found to be significant compared to the high efficiency of metal catalysts [119]. Indeed, although the conversion of DHA to MGO has been shown to be catalyzed by both Lewis and Brønsted acids, the conversion of MGO to LA appears to be catalyzed only by Lewis acids. It is, therefore, not surprising that LA, as a weak acid (pK_a_ = 3.86) [136], exhibits a poor catalytic effect on DHA to MGO conversion, and no catalytic effect on MGO to LA conversion [120].

Other products can be considered by a slight modification of this route. Performing the conversion reaction in methanol or ethanol instead of water produces methyl lactate and ethyl lactate, respectively (Figure 13). Ethyl lactate production by this process has received a lot of interest in recent years [131,137,138,139,140,141,142,143,144,145,146,147]. Methyl lactate is also of interest, as it is both a valuable chemical and an interesting alternative route to LA synthesis if processed further by simple hydrolysis [148,149,150]. The work of Morales et al. estimated methyl lactate production to be both energetically and economically more efficient than the one-step production process in water of LA due to lower energy demands of the different downstream separation and purification procedures [125].

Although MGO is capable of forming small polymeric structures by successive hydrations and reactions of MGO on itself (Figure 13) [90,151,152,153,154,155], it is worth noting that Nemet et al. showed that these oligomeric structures are only stable in the absence of water, and that, once these species are introduced into an aqueous solution, they revert to monomeric MGO [110,156].

Some papers have reported the production of acetol (monohydroxyacetone) subsequent to the production of MGO [157,158]. To our knowledge, no further evidence is presented in the literature to clarify the reaction pathway for acetol production; it could be produced directly from MGO as reported, but it could also come from earlier reaction intermediates, as both papers deal with reactions on hexose sugars systems, or from additional rearrangements after MGO production.

In summary, DHA solutions can lead to lactic acid formation as an undesirable by-product. This reaction has been studied to selectively convert DHA to lactic acid or alkyl lactates by a successive dehydration–rehydration reaction. This route for lactic acid synthesis has been estimated to be both economically and ecologically promising. A side-reaction may occur on the MGO intermediate in water-containing organic solvents, leading to complex, multi-hydrated oligomeric structures of MGO; but the equilibrium is easily reverted to MGO monohydrate in aqueous media.

### 3.5. Base-Catalyzed Condensation of Trioses

Condensation reactions have already been mentioned in a previous section, but a particular type of triose condensation reaction requires further attention. In alkaline DHA solutions, another type of product (a mixture of several hexoses) can be formed through base-catalyzed aldol condensation.

Fischer and Baer reported earlier the production of a mixture of hexoses, either from Gld alone or from a 1:1 mixture of DHA and Gld in alkaline conditions [159]. Regardless of the starting reagent, the process yielded a similar mixture, almost exclusively composed of sorbose and fructose, in a ratio depending on the starting trioses. Several studies followed the work of Fischer and Baer [160,161,162,163], which have been reviewed and enriched with complementary experiments by Gutsche et al., revealing the confusion in previous studies of sorbose with dendroketose [84]. Regarding these new elements, Gutsche et al. provided a general reaction scheme for the base-catalyzed condensation of trioses (Figure 14).

In the work of Gutsche et al., both Gld and DHA aldolizations in aqueous alkaline solutions have been confirmed to follow first-order kinetics with respect to the triose and the base. However, there is some evidence suggesting that the reaction requires water. Although pyridine is known to be a powerful catalyst of this reaction in aqueous solution, in anhydrous pyridine as a solvent, the reaction yielded no aldol condensation products [71,83,160]. In aqueous solutions, the condensation of trioses to hexoses appears to be catalyzed by pyridine and pyridine-like compounds, with evidence that the optimal catalytic activity correlates with a sterically unhindered pyridine nitrogen atom [84]. Gutsche et al. also concluded that the ionization step was rate-limiting in this reaction, and the rates of reaction following ionization were sufficiently faster than the rates of reaction in the reverse direction so that conversion of triose to hexose was essentially complete if treated with the base for a sufficiently long time. These elements could help to achieve the stereoselective synthesis of rare carbohydrates, for which they have already been used [164,165,166].

### 3.6. DHA-Based Innovative Polymers

In the previous section, the wide range of adverse reactions that DHA can undergo was presented. This section aims to provide the reader with several examples of the DHA-based polymers reported so far, which could help researchers to capitalize on the potential for DHA as an emerging bio-based synthon.

Most of the reports about DHA-based polymers deal with polycarbonates. The first synthesis of a polycarbonate containing DHA moieties was reported by Wang et al. in 2004 [167]. However, DHA is not involved anywhere in the synthesis route; the 1,3-dihydroxypropan-2-one pattern is obtained by multi-step synthesis involving the oxidation of the starting reagent diethyl malonate to ketomalonate [167,168]. Even if Wang and coworkers did not purposely proceed to the hydrolysis of their polymer to reveal the DHA pattern, they were already claiming its enhanced biodegradability by analogy with other aliphatic polycarbonates bearing functional groups.

The first synthesis of an actual DHA-based polycarbonate was reported in 2006 by Zelikin and Putnam [169]. The synthesis route goes through ketone protection, carbonation by cyclization of the diol, followed by ring-opening polymerization, and finally deprotection of the polymer by acid hydrolysis (Figure 15). The resulting material was shown to be reactive to amines by condensation on the ketone. It also showed unusual properties compared to other aliphatic polycarbonates, such as exceptional resistance to uniaxial compression and hydrophilic character, although it is insoluble in water. Several studies were then carried out, focusing either on the organo-catalyzed ROP of the DHA-derived carbonate (2,2-dimethoxytrimethylene carbonate, DHAC) [170], on reaction optimization [171], or on in vitro degradation tests [172]. A direct, one-step route to the aliphatic DHAC homopolymer, which avoids ketone protection by directly reacting DHA with dimethyl carbonate, organocatalyzed by 5%mol triazabicyclodecene (TBD), was reported by Banella (Figure 15) [173].

Numerous copolymers using DHAC have been reported (Figure 16). Diblock copolymers of PMPEG-*b*-PDHAC were used as a rapidly resorbable hemostatic biomaterial [174], or as drug delivery systems, their amphiphilic character leading to the formation of nanoparticles in aqueous solutions [175]. DHA-based polycarbonate hydrogel networks were assessed by Ricapito et al. and were shown to undergo unexpectedly rapid degradation in aqueous conditions, principally attributed to the presence of the DHA ketone group and short carbon chain [176]. Copolymers either of ε-caprolactone [170,177] or lactic acid [178,179,180] and DHAC were investigated as potentially bio-based polymers with enhanced biodegradability and biocompatibility. Hult et al. recently pioneered both alternating and random copolymers of DHAC with isosorbide (IS) [181], a promising bio-based monomer for sustainable alternative solutions to conventional, petrochemical-based commodity polymers [182,183,184]. Finally, Wang and co-workers reported an unusual copolymer composed of glycol-protected DHAC and 1,4-dioxane-2-one (DON), which showed interesting properties during in vitro drug controlled-release tests, with high hydrophilicity and easy degradation in aqueous media [185].

Other DHA-based polymers have also been reported. Banella investigated the one-step polycondensation of DHA with a range of aliphatic dimethyl diesters of various carbon chain length, ranging from C3 to C20, in an effort to develop new families of bio-based polyesters [173]. A study by Korley et al. looked at similar compounds, although synthesized using diacyl chlorides also in a one-step synthesis [186]. They reported biocompatibility in line with other prominently used implantable polyesters, such as PLA and PLGA, confirming the potential of this new class of polyesters for use as biomaterials in implantable devices, for example.

DHA has also found some interesting uses in polyurethanes, either as a crosslinker in waterborne polyurethanes [187,188], or by reaction of the alcohol functions of DHA with isocyanate-terminated prepolymers, such as MPEG-isocyanate [26]. However, except for these few recent examples, DHA is still rarely exploited in polyurethanes, which leaves the field open for future innovative work.

Finally, DHA has been reported to form highly complex and rare structures called spiroacetals, and their double-stranded polymeric derivatives, polyspiroacetals [189]. While no application was suggested for pure DHA polyspiroacetals, DHA-based spiroacetal moieties were successfully incorporated in aliphatic polycarbonates, resulting in increased Tg (42–61 °C) compared to aliphatic polycarbonates (−15 °C to 20 °C), although not reaching those of aromatic polycarbonates (≈150 °C for bisphenol A polycarbonates) [169]. However, surprisingly, the thermal stabilities of these DHA-dimer polymers approach that of the aromatic polycarbonates (>300 °C). The reported polyspiroacetals also showed evidence of low cytotoxicity, suggesting the need for a full biocompatibility evaluation [169]. These properties are desirable for high-strength applications in tissue engineering, matrices for controlled drug delivery, and patterned surfaces for cell growth, but additional studies are needed to explore further the potential of these compounds.

## 4. Conclusions

Dihydroxyacetone is one of the simplest carbohydrates and is a multifunctional compound bearing a functional group on each of its three carbon atoms. As it is naturally present in most living organisms, it is a promising compound for future innovative and sustainable materials, both for biocompatible and biodegradable polymers. The industrial supply of DHA is already ensured in a sustainable way, since production is carried out by microbial fermentation and processed in aqueous media, possibly using low environmental impact catalysts, such as enzymes or free cell extracts, among others. Research is ongoing to find a more efficient process, such as electrooxidation or aerobic oxidation, which would be easier and less expensive to implement due to the abundance of oxidizing agents involved (water and air, respectively). However, the excessive cost of the noble metals used as catalysts in both cases, as well as the lack of experience with these processes, limits their development at the industrial scale in the short term.

While several examples of DHA-based polymers have been mentioned previously, there are still many possibilities for the use of DHA as a bio-based building block. In this context, its high reactivity may represent a significant technical hurdle that may be difficult to overcome. DHA can undergo several types of reactions under mild conditions: isomerization, condensation and polymerization reactions, among others. The main challenge in performing a reaction involving DHA is not to improve the reactivity of the molecule, but rather seems to be the difficulty of stopping the reaction at a desired product due to several side-reactions. It is also possible to make use of some of these side-reactions to achieve the synthesis of products which would actually be the compounds of interest. An example is the use of LdB–AvE transformation to achieve the synthesis of lactic acid or alkyl lactates, a route estimated to be both energetically and economically more efficient than the current industrial process. Furthermore, there is still much room to further explore and understand the chemistry of DHA. Recent advances in green chemistry could be used in the exploration of easier conditions for handling DHA, for example, the use of innovative solvents such as ionic liquids or deep eutectic solvents as greener alternatives.

DHA is mainly produced via waste valorization from the biodiesel industry. It is readily available, making it a reagent of choice for the development of future sustainable materials. The financial argument for a low-cost, highly functional bio-based building block may encourage scientists to further explore the potential of DHA. We hope that researchers will find in this review inspiration and cautions to bear in mind for future research.

## Figures and Tables

**Figure 1 molecules-28-02724-f001:**
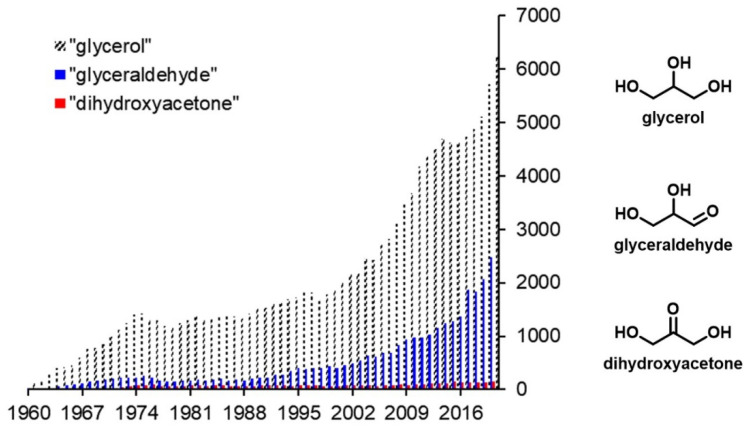
Numbers of publications by year from 1960 to 2021 for dihydroxyacetone and its closest derivatives. Data from Scopus extracted on 4 August 2022.

**Figure 2 molecules-28-02724-f002:**
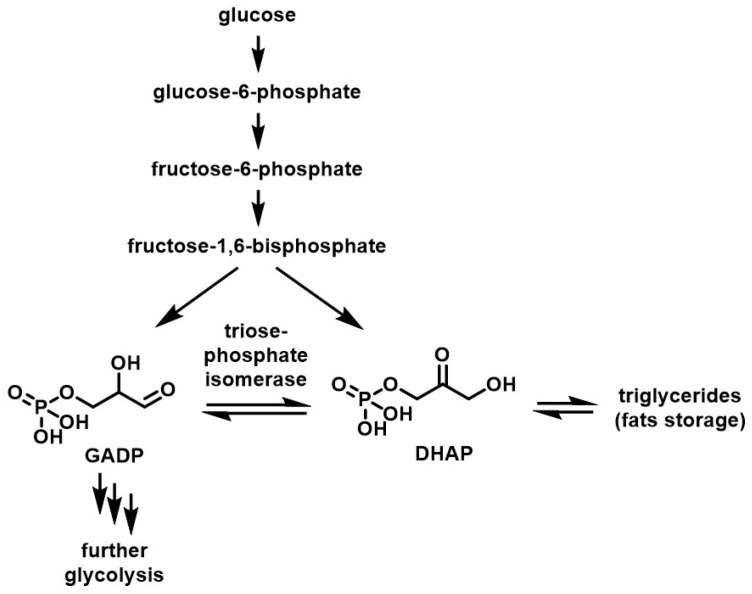
Simplified reaction path of the first steps of glycolysis.

**Figure 3 molecules-28-02724-f003:**
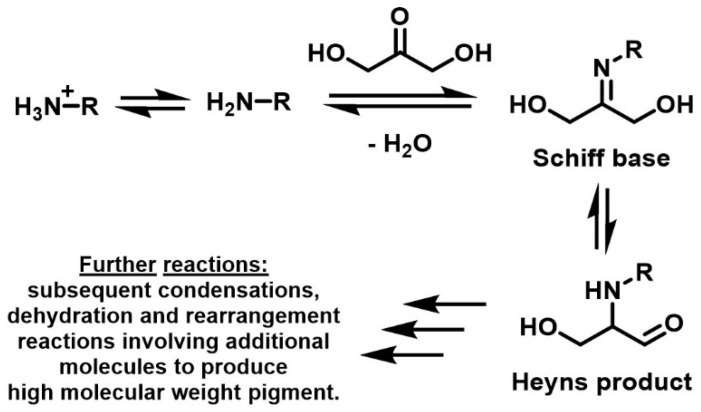
Reactions of DHA with amino groups leading to pigment formation (melanoidins). Adapted from [21] with permission from Elsevier.

**Figure 4 molecules-28-02724-f004:**
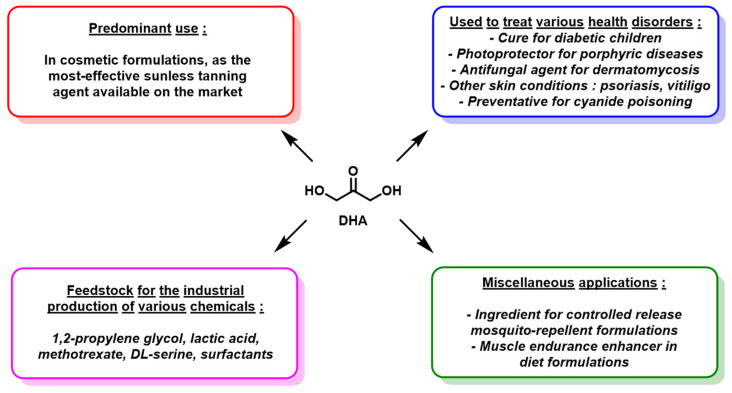
Synoptic representation of the reported applications of raw DHA.

**Figure 5 molecules-28-02724-f005:**
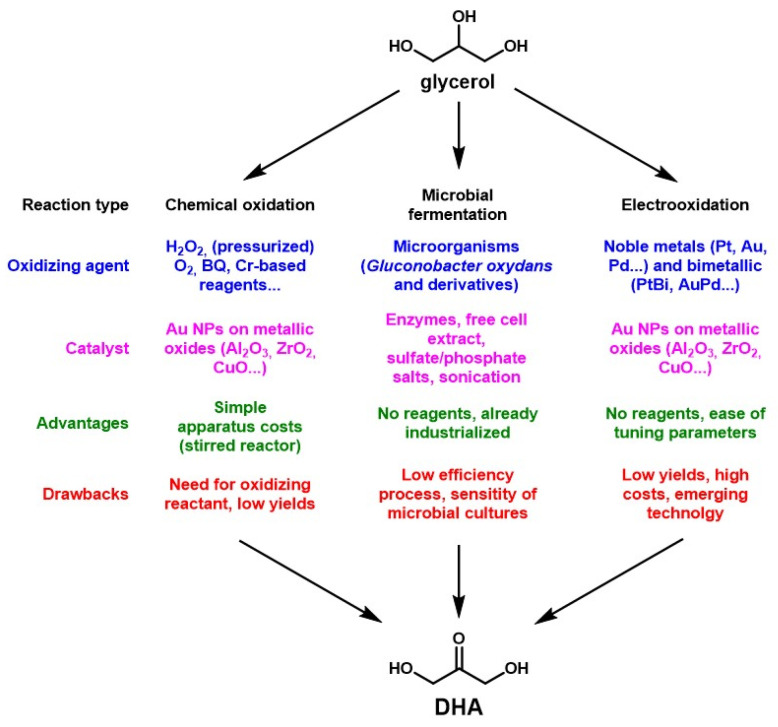
Schematic resume of the reported synthesis routes to produce DHA from glycerol.

**Figure 6 molecules-28-02724-f006:**
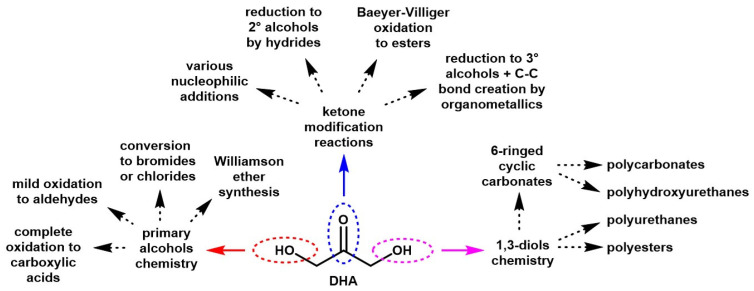
Scheme of the reactive sites of DHA and potential reactions.

**Figure 7 molecules-28-02724-f007:**
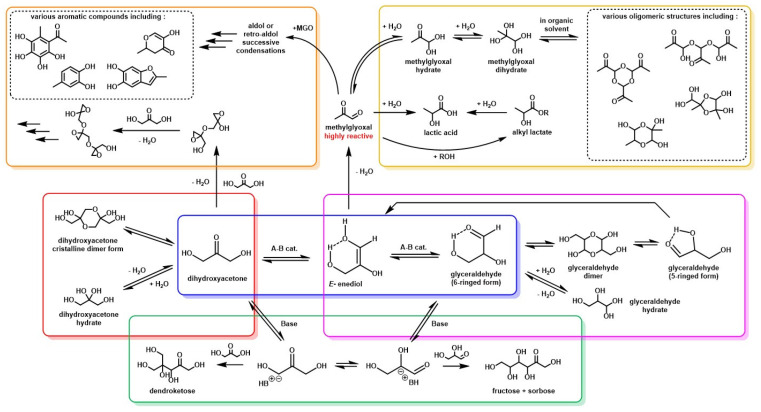
General reaction scheme of the various compounds derivable from DHA in common solutions.

**Figure 8 molecules-28-02724-f008:**
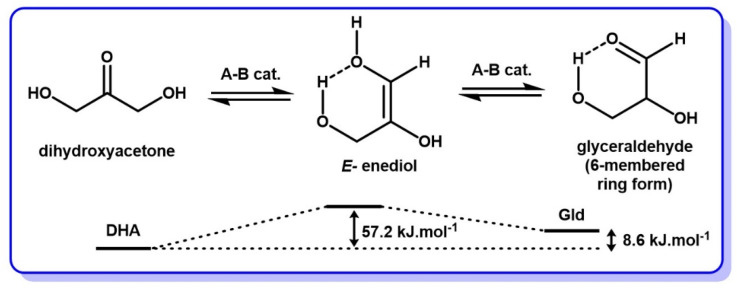
DHA–Gld interconversion mechanism through an enediol intermediate. Energy diagram obtained with B3LYP/6-31G* model, adapted with permission of the Royal Society of Chemistry from [7]; permission conveyed through Copyright Clearance Center, Inc.

**Figure 9 molecules-28-02724-f009:**
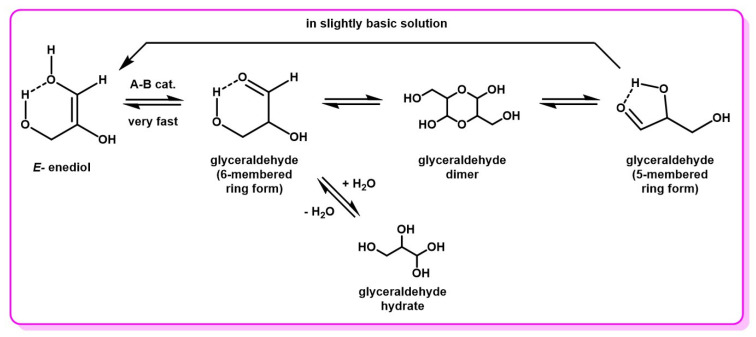
Different forms of glyceraldehyde in aqueous solution.

**Figure 10 molecules-28-02724-f010:**
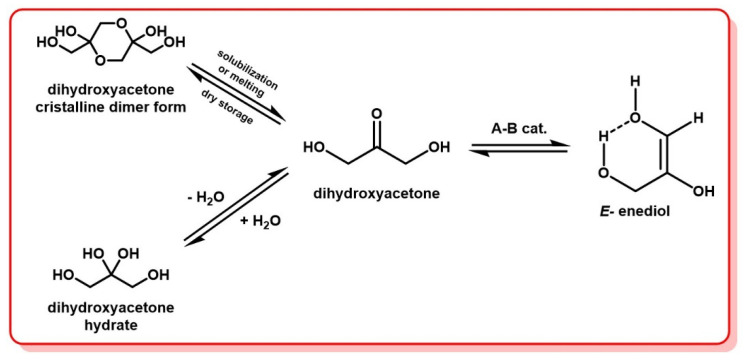
Common forms of DHA and interconversion to enediol intermediate in aqueous solution.

**Figure 11 molecules-28-02724-f011:**
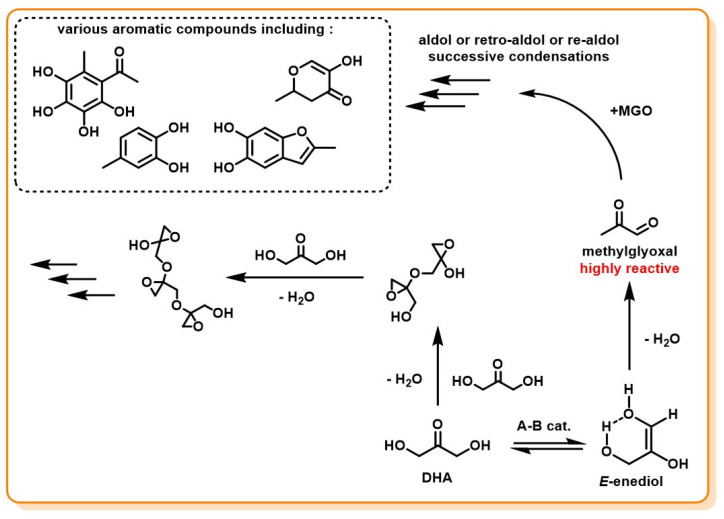
Condensation reaction paths from DHA and the enediol intermediate.

**Figure 12 molecules-28-02724-f012:**
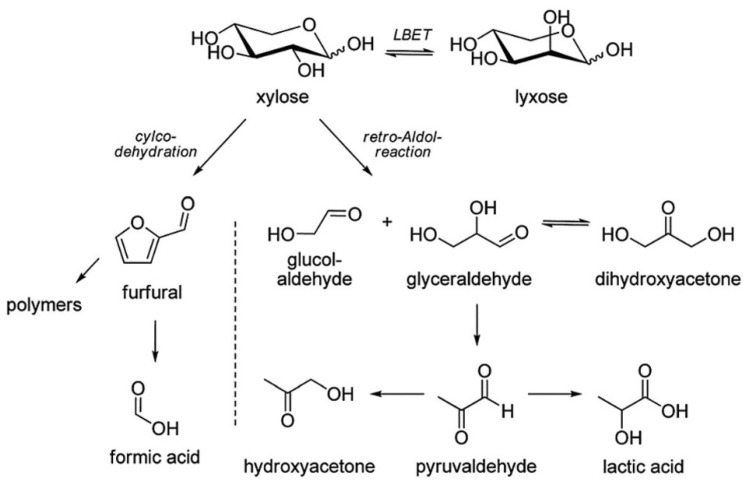
Reaction scheme of xylose degradation and the main degradation products formed. Adapted with permission of the Royal Society of Chemistry from [106]; permission conveyed through Copyright Clearance Center, Inc.

**Figure 13 molecules-28-02724-f013:**
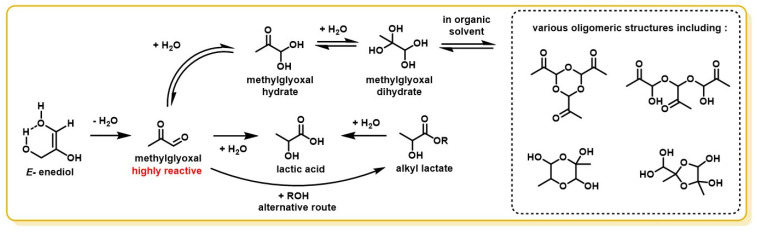
MGO hydrates and lactic acid conversion from the enediol intermediate of DHA–Gld interconversion.

**Figure 14 molecules-28-02724-f014:**
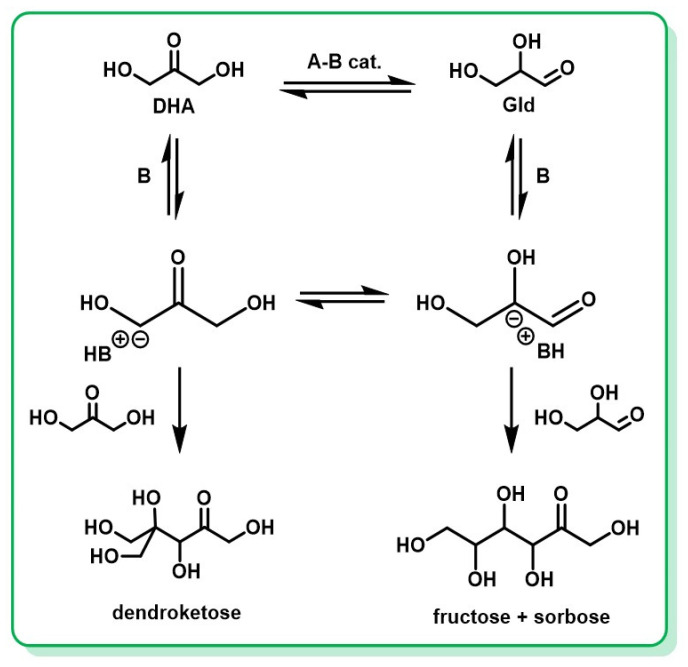
Reaction scheme of the base-catalyzed condensation of trioses. Adapted from [84].

**Figure 15 molecules-28-02724-f015:**
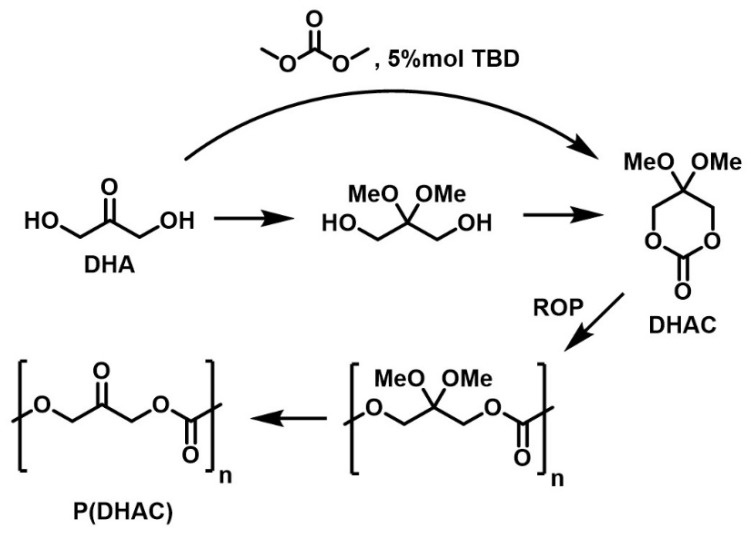
Routes to DHAC synthesis and polymerization to P(DHAC).

**Figure 16 molecules-28-02724-f016:**
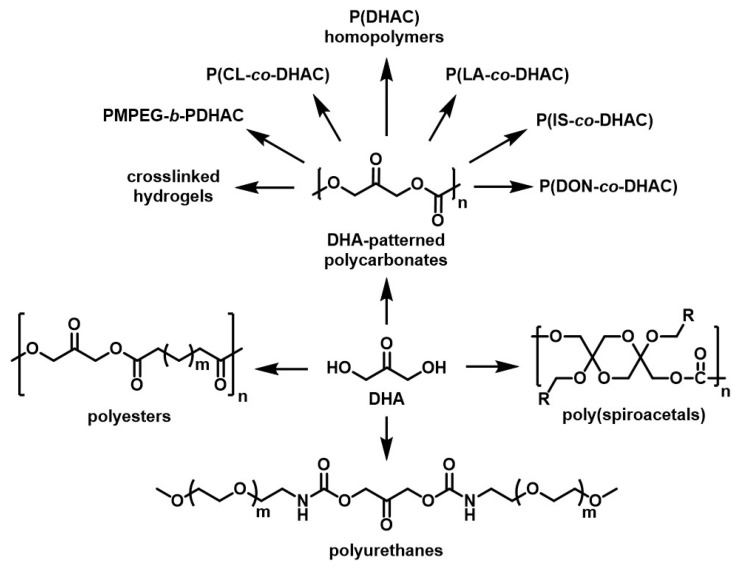
Scheme of the various DHA-based polymers reported so far.

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
