# Peer review of "Dihydroxyacetone: A User Guide for a Challenging Bio-Based Synthon"

_molecules, 2023, doi:10.3390/molecules28062724_

Round 1
Reviewer 1 Report
The authors decscribe the chemistry and synthesis route of dihydroxyacetone. The review is well written and very clear to read. I would recommend to add a section with a schematic representation of the current applications of this molecule and the possible future applications, especially in light of the recent developments in terms of polymerization/copolymerization.
Author Response
We thank the reviewer for his/her comments.
We have added a synoptic scheme of the current applications of DHA (Figure 4) as per his/her recommendation.
Reviewer 2 Report
This is a quite complete review article, but in contrast to the aim of the article to focus on DHA based polymers in the third section, this section concentrates more on general synthetic possibilities but seems to be rather short on "innovative polymers" which would earn more consideration. So subchapter 3.6 should be expanded, may be as a 4th section before the conclusions.
But, apart from this constraints, the review is well written, informative and could be printed as it is.
Author Response
We thank the reviewer for his/her comments.
The section dedicated to DHA-Based polymers is indeed rather short due to the lack of such polymers. We do believe that DHA is a promising building block for materials ans this is what we are trying to convey in the present review. However, the rich and somewhat difficult to control chemistry of DHA is likely responsible for the still underdevelopped use of DHA in polymer materials. As a result we could not really exapand the section on innovative DHA-based polymers. we have, however, reworded the description of this section in the introduction.
Reviewer 3 Report
The author has succeeded in writing a useful review of DHA for the efficient use of glycerol, a main byproduct of FAME. This manuscript provides comprehensive and timely information about the preparation, reaction and application of DHA and may be useful for scientists and managers working in green material chemistry. I think this manuscript is acceptable in the present form.
Author Response
We thank the reviewer for his/her comments.
Round 2
Reviewer 2 Report
No further comments